# Resonance Scanning as an Efficiency Enhancer for EEG-Guided Adaptive Neurostimulation

**DOI:** 10.3390/life13030620

**Published:** 2023-02-23

**Authors:** Alexander I. Fedotchev, Sergey B. Parin, Sofia A. Polevaya

**Affiliations:** 1Institute of Cell Biophysics, Russian Academy of Sciences, 3 Institutskaya St., Pushchino, 142290 Moscow, Russia; 2Department of Psychophysiology, Lobachevsky State University of Nizhni Novgorod, 23 Prospekt Gagarina, 603950 Nizhny Novgorod, Russia

**Keywords:** closed-loop stimulation, electroencephalogram (EEG), rhythmical EEG components, on-line modulation, resonance scanning, cognitive improvement, stress management

## Abstract

Electroencephalogram (EEG)-guided adaptive neurostimulation is an innovative kind of non-invasive closed-loop brain stimulation technique that uses audio–visual stimulation on-line modulated by rhythmical EEG components of the individual. However, the opportunity to enhance its effectiveness is a challenging task and needs further investigation. The present study aims to experimentally test whether it is possible to increase the efficiency of EEG-guided adaptive neurostimulation by pre- strengthening the modulating factor (subject’s EEG) through the procedure of resonance scanning, i.e., LED photostimulation with the frequency gradually increasing in the range of main EEG rhythms (4–20 Hz). Thirty-six university students in a state of exam stress were randomly assigned to two matched groups. One group was presented with the EEG-guided adaptive neurostimulation alone, whereas another matched group was presented with the combination of resonance scanning and EEG-guided adaptive neurostimulation. The changes in psychophysiological indicators after stimulation relative to the initial level were used. Although both types of stimulation led to an increase in the power of EEG rhythms, accompanied by a decrease in the number of errors in the word recognition test and a decrease in the degree of emotional maladjustment, these changes reached the level of significance only in experiments with preliminary resonance scanning. Resonance scanning increases the brain’s responsiveness to subsequent EEG-guided adaptive neurostimulation, acting as a tool to enhance its efficiency. The results obtained clearly indicate that the combination of resonance scanning and EEG-guided adaptive neurostimulation is an effective way to reach the signs of cognitive improvement in stressed individuals.

## 1. Introduction

Creation and improvement of closed-loop techniques for non-invasive brain stimulation is an exciting and rapidly developing field of neuroscience. These technologies are known to modulate brain circuits and networks for successful treatment of affective symptoms and disorders [1] and to improve human performance in normal individuals [2]. EEG-guided adaptive neurostimulation is an innovative kind of such a technique that uses sensory stimulation on-line modulated by rhythmical EEG components of the individual. This technique has a number of advantages, such as high personalization of the treatment due to the use of feedback from the subject’s own bioelectrical characteristics, automatic operation without conscious efforts of an individual, and integration of interoceptive signals in the regulatory mechanisms of the brain [3].

EEG-guided adaptive neurostimulation has been successfully used for the cognitive rehabilitation of stroke patients [4] and for the correction of stress-induced states [5]. The developed technique includes simultaneous presentations of music-like stimuli on-line controlled by the feedback from discrete low-frequency (4–20 Hz) components of the subject’s electroencephalogram (EEG) oscillators, and presentations of rhythmic light stimuli generated from the raw EEG of the subject. It was shown that under such bisensory stimulation a decrease in stress level, normalization of the EEG, and positive shifts in the psycho-emotional status of human participants are observed due to the optimal conditions for involving the integrative, adaptive, and resonance mechanisms of the central nervous system into human functional state regulation [5]. However, the opportunity to enhance the effectiveness of EEG-guided adaptive neurostimulation is a challenging task and needs further investigation, especially in the stress-related line of research. In particular, it is critically important to effectively correct the consequences of stress, which has detrimental effects on the psychological well-being, cognitive function, and physiological health of university students [6] and is inversely related with mindfulness and with the results of the most challenging exam [7].

The effects of the described type of adaptive neurostimulation significantly depend on the strength of the modulating factor, that is, the EEG of a person. Therefore, one of the possible ways to increase the effectiveness of EEG-guided adaptive neurostimulation may be the preliminary activation of low-frequency rhythmic EEG components of the individual. With this aim, the procedure of resonance activation of the EEG could be used. It consists of LED photostimulation with a stepwise increase in frequency in the range of theta, alpha, and low beta EEG rhythms, i.e., 4–20 Hz [8]. These EEG rhythms are known to represent several mental conditions, such as deep and wakeful relaxation, good mood, calmness, increased self-awareness and focus, learning of new information, and active attention [9]. Therefore, resonance activation of these rhythms is expected to induce the signs of cognitive rehabilitation of a person. Recently, this procedure was successfully used for resonance activation of cortical rhythms in younger school children [10], where it was named the “resonance scanning” technique.

The aim of the present study is to test, experimentally, whether the combination of resonance scanning with EEG-guided adaptive neurostimulation is more effective in the stress management and cognitive improvement of stressed students than the EEG-guided adaptive neurostimulation alone. The experiments involved two matched groups of university students in a state of exam stress. After registration of background psychophysiological parameters, one group was presented with the EEG-guided adaptive neurostimulation alone, whereas another group was presented with the combination of resonance scanning and EEG-guided adaptive neurostimulation. To compare the effects of both types of stimulation, the changes in psychophysiological indicators after stimulation relative to the initial level were used.

## 2. Materials and Methods

### 2.1. Participants

The experiments involved 36 students from Lobachevsky State University of Nizhni Novgorod. Study participants met the following inclusion criteria: healthy university students under exam stress, passed 3 or 4 years of study. Exclusion criteria were: no subjects had taken any medications for the nervous system, or had a history of neurological or mental diseases. The students were randomly assigned to two groups matched by number, gender, age, and study background of participants (Table 1).

According to initial interview, the students were in a state of exam stress and voluntarily agreed to participate in the experiment. The study was carried out in accordance with the Declaration of Helsinki (2013) and with ethical principles established by the European Convention for the Protection of Vertebrata used for Experimental and other Scientific Purposes (the Convention was passed in Strasburg, 18 March 1986, adopted in Strasburg, 15 June 2006). The study was approved by the Ethics Committee of the Lobachevski State University of Nizhny Novgorod. Each participant provided written informed consent form.

### 2.2. Questionnaire

In order to assess the initial psychophysiological state of the subjects, at the beginning of each experiment they were interviewed and psychologically tested using two tests: the “ED” test [11] and the semantic attention test using word recognition [12].

In the “ED” test, the participants were presented with a circle, divided into four sectors. Within each sector, there were adjectives describing the emotional state of a person, corresponding to the four basic requirements of personality: security, independence, achievement, unity–intimacy. Participants were asked for three attempts to choose the sector in the circle corresponding to their current condition. Each sector had its own score (from 1 to 4), which was not shown to the participant. The scored sum of points served as an indicator of the degree of emotional maladjustment of a person.

In the word recognition test, a set of words (21 in total) was used, in which seven (30%) were target words. It was proposed for the subject to memorize seven target words and press the mouse button when the target word appeared on the screen. The total number of errors (number of missed target stimuli plus number of “false alarms”—pressing when a neutral word appears) served as indicator to assess the person’s cognitive and memory functions.

### 2.3. Procedure

After psychological testing, the subjects were installed with Ag/AgCl electrodes (one active and two reference), stereo headphones (Philips SBC HL140), and tinted glasses having embedded red LEDs with maximal power of 100 μW. The subjects were not given any task but were asked to sit quietly with their eyes closed during the entire examination. Then, the EEG was recorded to identify the subject’s alpha EEG oscillator and to control the stimulation parameters. At the end of each experiment, the subjects were retested and interviewed about their feelings during the treatment.

### 2.4. EEG Recording and Analysis

EEG recording was monopolar, with the use of an active electrode at point Cz (International system 10/20) and ear electrodes as a reference and ground. EEGs were amplified and digitized using a multichannel EEG amplifier Brainsys (Hardsoft, Russia). The sampling rate of EEG data was set at 128 Hz; the data was band-pass filtered with a 0.5–60 Hz filter to reduce ultra-noise. During resting-state EEG registration, the dominant narrow frequency (0.6 Hz) EEG oscillator from the alpha (8–13 Hz) EEG band was identified in each participant. For this purpose, procedures of fast Fourier transform were performed for short (5 s) periods of resting-state EEG, which were sequentially shifted relative to each other with 50% overlap. When individual spectral peaks are sequentially accumulated for a period of resting-state EEG recording, the resulting spectrum is based on the summation of many short-term spectra, it has 0.2 Hz frequency resolution, and provides information on the narrow-band EEG oscillator (peak frequency ± 0.2 Hz), stable for the person. The revealed alpha EEG oscillator of the individual was then used for automatic modulation of acoustic stimuli.

### 2.5. Resonance Scanning

In the experiments with preliminary resonance scanning, LED stimulation was presented according to a specially developed program. It was a series of flashes with stepwise increasing frequency in the range from 4.0 to 20.0 Hz in increments of 0.2 Hz. The duration of stimulation at each fixed-frequency step was 3 s, and the total duration of photostimulation was 240 s.

### 2.6. Adaptive Neurostimulation

In the case of adaptive neurostimulation, the subjects were presented with music-like signals resembling the sounds of a flute in timbre, smoothly changing in height and intensity in direct proportion to the current amplitude of the person’s alpha EEG oscillator. Simultaneously, the participants were presented with rhythmic light stimulation controlled by the subject’s total EEG. This was carried out by normalizing the digitized EEG values, in which the largest negative value of the EEG signal corresponded to the minimum, and the largest positive value corresponded to the maximum luminescence of the LEDs. EEG recording was continued after the end of stimulation to identify the aftereffects.

### 2.7. Data Analysis

A statistical data analysis was performed by using the software package Sigma-Plot 11.0. To check the normality of the data distribution, the Shapiro–Wilk W-test was used, since this test has the highest power and is most preferable for small samples. The analyzed distribution was assessed as normal at the level of statistical significance *p* > 0.05. In the case of normal distribution of data for descriptive statistics, the mean (M) and standard errors (m) were used; to assess the significance of differences, Student’s *t*-test was used. In the case of a distribution other than normal, the median (Me) and the range of values from the first (Q1) to the third (Q3) quartile were used for descriptive statistics of features, and the Mann–Whitney test was used to assess the significance of differences. The critical level of statistical significance was taken as *p* < 0.05. Repeated measures multiway ANOVA was used to assess the statistical significance of the shifts for each psychophysiological indicator under stimulation relative to the background. Power shifts for theta, alpha, beta EEG rhythms, changes in a number of errors in the word recognition test, and the shifts of the level of emotional maladjustment were processed.

## 3. Results

In order to observe visually the spectral dynamics of electrophysiological changes during each experiment, the stimulation spectra and the EEG spectra were sequentially calculated and compared. A typical example of such spectral dynamics for one of the subjects from the group with resonance scanning is shown in Figure 1.

*X*-axis—the frequency of the spectrum, Hz;

*Y*-axis—the time of the experiment, s;

*Z*-axis is the spectral density (µV/Hz) reflected in the level of brightness.

In Figure 1, one can see that after a 120 s background recording, the resonance scanning begins. As the stimulation frequency increases, resonant spectral peaks are observed in the EEG spectra, coinciding exactly in frequency with the current stimulation frequency (resonant EEG reactions at the stimulation frequency). In addition, resonant spectral peaks are also observed at the frequency of the second harmonic of stimulation (rhythm multiplication EEG reaction). These spectral peaks form two slanting lines on the right side of the figure, reflecting the resonant activation of the EEG at the stimulation frequency and its harmonic. After a short pause, adaptive neurostimulation begins. The local resonance peaks on the spectral EEG curves resemble the spectral peaks recorded for the dynamics of stimulation. One can also see the growth of spectral EEG amplitudes under adaptive neurostimulation relative to the background.

A typical example of spectral dynamics for a subject from the group without resonance scanning is shown in Figure 2.

Figure 2 shows that after a 120 s background recording, the adaptive neurostimulation begins, where the local resonance peaks on the spectral EEG curves resemble the spectral peaks recorded for the dynamics of stimulation. The local resonance peaks are especially pronounced for the EEG oscillator (approximately 11.0 Hz) in the alpha range. After stimulation completion, the spectral EEG curves are similar to the background ones.

In order to quantitatively assess the EEG effects of resonance scanning, the powers of EEG rhythms in the background and during photostimulation were compared (Figure 3).

In Figure 3, one can see that during the procedure of resonance scanning, a significant increase relative to background is observed for all powers of EEG rhythms.

To evaluate the effects obtained as a result of both types of stimulation, the psychophysiological indicators before and after treatment in the experiments with vs. without resonance scanning were compared (Table 2).

The data from Table 2 show that in both experimental sessions the growth of power values of EEG rhythms is observed after stimulation. However, significant power increments for alpha and beta EEG rhythms are registered only in the experiments with resonance scanning. These EEG effects in the experiments with resonance scanning are accompanied by a significant decrease in the number of errors in the word recognition test and a significant decrease in the level of emotional maladjustment.

Interviewing the subjects about their subjective feelings during the experiments revealed that all subjects reported a decrease in stress level and an improvement of own emotional state as a result of their participation in the experiments. After stimulation, the participants reported to feel more relaxed, alert, with a greater sense of well-being. Most of the subjects estimated the performed procedures as pleasant and calming, especially the experiments with resonance scanning.

## 4. Discussion

Our results support the proposition that the combination of resonance scanning with EEG-guided adaptive neurostimulation is more effective in the cognitive improvement of stressed students than the EEG-guided adaptive neurostimulation alone. Although both groups of subjects (with or without preliminary resonance scanning) showed an increase in EEG powers, decrease in errors in the word recognition test, and decrease in emotional maladjustment, these changes only reached significance in the group with preliminary resonance scanning. Thus, resonance scanning could be used as a tool to enhance the efficiency of EEG-guided adaptive neurostimulation in stress-reducing and cognitive-enhancing procedures.

Resonance scanning makes it possible to observe the dynamics of resonant EEG responses at the stimulation frequency (rhythm entrainment) and at the frequency of harmonics (rhythm multiplication). From the literature, it is known that these EEG responses reflect the degree of functional lability, adaptive potential, and non-linearity of brain reactions due to the mechanisms of interaction between endogenous and exogenous oscillations [13], neural entrainment [14,15], and controlled neuroplasticity [16,17]. Based on these mechanisms, resonance scanning increases the brain’s responsiveness to subsequent adaptive neurostimulation, acting as a kind of pre-tuning of the brain, which cause the activation of potential resonators in the EEG spectrum.

In the recent literature, several kinds of varying frequency sensory stimulation to induce neuroplasticity and cortex responsivity are described. For example, trains of quickly repeated auditory or visual stimuli, i.e., tetanic stimulation is proposed to alter the EEG in humans [18]. Short series of light inputs over a broad frequency range (“chirp” stimulation) is designed to uncover the features of visual cortex responsivity [19]. To induce neural plasticity in humans, quadripulse stimulation is developed, where one stimulation burst consisting of four monophasic pulses is given every 5 s for 30 min [20]. In comparison with these kinds of stimulation, our resonance scanning technique has obvious advantages, such as its dynamic nature, software-controlled digital parameters of stimulation, and an opportunity to reveal and activate a fine structure of the individual EEG spectrum.

Due to the implementation of resonance scanning before EEG-guided adaptive neurostimulation, a significant increase in power for alpha and beta EEG rhythms was observed together with positive results of cognitive testing. Similar results were recently obtained using musical neurofeedback in healthy subjects [21]. However, the authors achieved successful alpha wave induction and subsequent improvements in cognitive functions after a long period of training. Similar positive effects were achieved in our study after only a single treatment procedure. This advantage permits us to consider a combination of resonance scanning with EEG-guided adaptive neurostimulation as a newly developed kind of oscillotherapeutics [22,23] that can effectively reduce stress and quickly produce the signs of cognitive rehabilitation in persons under exam stress via identifying neuronal network oscillations as potential therapeutic targets.

It is important to note that the developed combination of resonance scanning with EEG-guided adaptive neurostimulation is in line with several promising trends in the development of non-invasive brain stimulation technologies. One of them is a quickly arising line of research related to development of the closed-loop brain state-dependent stimulation (CLBSDS) technique for modulating brain circuits and networks in clinic and to improve mental health and well-being in normal people [1,24]. Another trend is the rapidly evolving field of physiologically informed neuromodulation methodology and techniques, dealing with stimulation paradigms that are guided by patients’ neurophysiology to improve therapy efficacy and make patient management more efficient [25]. Furthermore, our data are in line with the results of a recent study, which demonstrate that a dysregulation of brainwaves may account for symptoms and affective disorders, and that they can be partially relieved by audio–visual entrainment [26].

The present study also has some limitations, such as the small number of participants and the lack of using special stress tests. We used a comparison of the data of the initial and final survey of the subjects for the evaluation of the level of stress. All subjects initially declared the presence of exam stress, based on their own subjective assessments, and after experimental procedures, they reported a decrease in stress level and an improvement in their emotional state and mood. These limitations will be eliminated in future studies. However, these limitations do not affect the reliability of the conclusions drawn. 

## 5. Conclusions

Our study clearly indicates that preliminary resonance scanning causes an activation of potential resonators in the EEG spectrum and increases the brain’s responsiveness to subsequent EEG-guided adaptive neurostimulation. As a result of such a combination of exogenous and endogenous rhythmical stimulations, the significant stress-reducing and cognitive improvement outcomes are registered after only a single treatment procedure due to the progressive involvement of the resonant and integrative mechanisms of the central nervous system and the mechanisms of neuroplasticity. The developed combined (resonance scanning plus EEG-guided adaptive neurostimulation) technique is non-invasive and highly personalized since it is based on the feedback from the subject’s own bioelectrical processes. In addition, it is characterized by automatic operation without conscious efforts of an individual, which provides the opportunity to correct unfavorable shifts of functional state in patients with altered levels of consciousness, elderly people, and children. In general, our research shows that the developed combined neurostimulation approach after the thorough elaboration could be used in a wide range of cognitive improvement interventions for specialists in extreme professions, in psychological relief rooms at work, in educational institutions to enhance human cognitive activity and learning processes, as well as in scientific research.

## Figures and Tables

**Figure 1 life-13-00620-f001:**
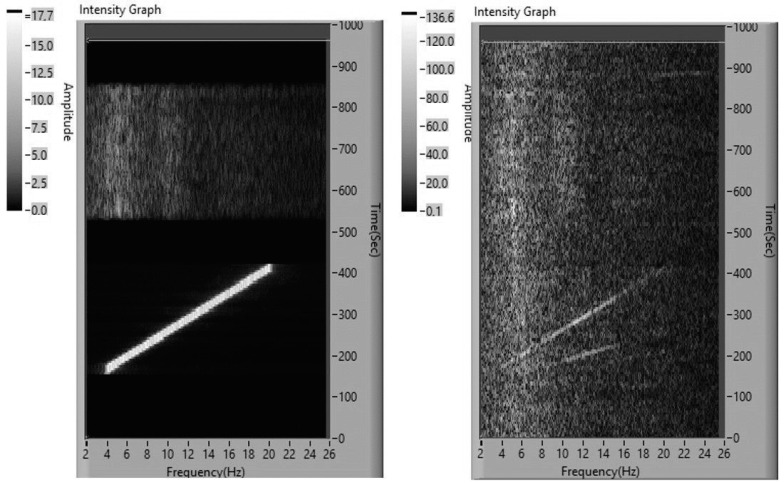
Dynamics of the stimulation (**left**) and the EEG (**right**) spectra during the experiment with subject R08.

**Figure 2 life-13-00620-f002:**
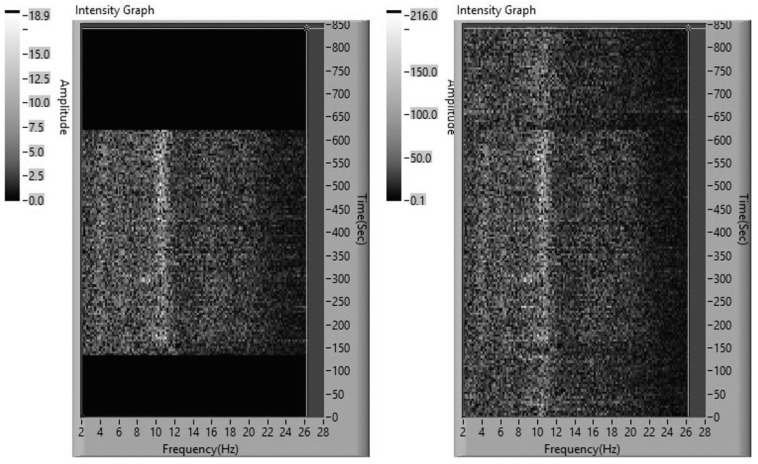
Dynamics of the stimulation (**left**) and the EEG (**right**) spectra during the experiment with subject E12. Designations as in Figure 1.

**Figure 3 life-13-00620-f003:**
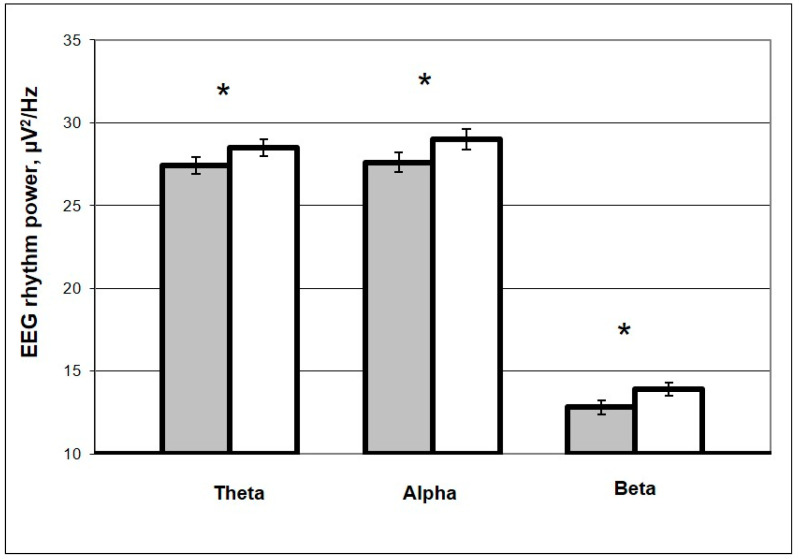
EEG theta-, alpha-, beta-rhythm powers before (dark columns) and during (light columns) resonance scanning. Note: *—*p* < 0.05 (Mann–Whitney test).

**Table 1 life-13-00620-t001:** Composition of two groups of university students that participated in the experiments with or without resonance scanning.

Group Attribute	Group	
	With Resonance Scanning	Without Resonance Scanning
Number of participants	18	18
Gender (females/males)	11/7	11/7
Age (years)	20.7 ± 0.2	21.0 ± 0.2
Years of study	3–4	3–4

**Table 2 life-13-00620-t002:** Objective and subjective indicators before and after stimulation and their levels of difference *p* in the experiments with vs. without resonance scanning (RS).

Indicators	Experimental Session	Before Stimulation	After Stimulation	*p*
Theta EEG power (µV^2^/Hz)	With RS	27.3 ± 1.4	27.8 ± 1.4	0.194
	Without RS	28.2 ± 1.4	28.6 ± 1.4	0.166
Alpha EEG power (µV^2^/Hz)	With RS	**28.3** **± 1.9**	**31.9** **± 2.1**	**0.001**
	Without RS	28.0 ± 1.8	28.6 ± 1.7	0.095
Beta EEG power (µV^2^/Hz)	With RS	**12.8** **± 0.9**	**13.7** **± 0.8**	**0.007**
	Without RS	14.3 ± 0.9	14.5 ± 1.0	0.361
Errors in word recognition test (digits)	With RS	**4.4** **± 0.5**	**2.5** **± 0.3**	**0.002**
	Without RS	4.2 ± 0.4	3.6 ± 0.5	0.207
Level of emotional maladjustment (scores)	With RS	**4.6** **± 0.4**	**2.8** **± 0.3**	**0.008**
	Without RS	4.3 ± 0.8	3.8 ± 0.8	0.571

Note: the values of changes with the significance level *p* < 0.05 are shown in bold (Mann–Whitney test).

## Data Availability

Web-platform COGNITOM (cogni-nn.ru).

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
