# Peer review of "Resonance Scanning as an Efficiency Enhancer for EEG-Guided Adaptive Neurostimulation"

_life, 2023, doi:10.3390/life13030620_

Round 1
Reviewer 1 Report (Previous Reviewer 1)
Authors made important corrections and improvements to their manuscript and it is now ready for publishing.
Author Response
Dear Reviewer 1,
Thank you very much for your efforts, helpful comments and final decision.
Sincerely – Alexander Fedotchev & co-authors.

Reviewer 2 Report (Previous Reviewer 2)
The changes in the paper are acceptable.
Author Response
Dear Reviewer 2,
Thank you very much for your efforts, helpful comments and final decision.
Sincerely – Alexander Fedotchev & co-authors.

Reviewer 3 Report (Previous Reviewer 3)
I appreciate that the authors have addressed and answered my previous comments. However, after reading the current submission, I still feel confused regarding the terminology. I don't know why the authors prefer using "Cognitive rehab" throughout the paper - they enroll young normal subjects with normal cognitive functioning, and the subjects do not need any "rehab". On the other, the authors used "stress management" as a potential implication. However, it is not clear how the authors assess the level of mental stress of the subjects at baseline. "Under exam stress" is a very broad and subjective criterion and the authors should use a quantifiable measure to better define the stress level.
Author Response
I appreciate that the authors have addressed and answered my previous comments. However, after reading the current submission, I still feel confused regarding the terminology. I don't know why the authors prefer using "Cognitive rehab" throughout the paper - they enroll young normal subjects with normal cognitive functioning, and the subjects do not need any "rehab". On the other, the authors used "stress management" as a potential implication. However, it is not clear how the authors assess the level of mental stress of the subjects at baseline. "Under exam stress" is a very broad and subjective criterion and the authors should use a quantifiable measure to better define the stress level.
Dear Reviewer 3,
Thank you very much for your efforts and helpful comments. We agree that the term “cognitive rehabilitation” is not applicable for young students with normal cognitive functioning. Sorry, in previous version of the manuscript we did not correct this term throughout the whole paper. In the current submission, we additionally corrected the term “cognitive rehabilitation” to “the signs of cognitive improvement/enhancement” throughout the whole paper (lines 18-19, 278, 334, 345).
We agree that the lack of using special stress tests is the main limitation of our study. In further research, we plan to develop a 10-point visual analog scaling technique in order to quantify the initial and final levels of stress.
We have added some fragments to the section “limitations” of Discussion as follows:
The present study also has some limitations, such as small number of participants and the lack of using special stress tests. We used a comparison of the data of the initial and final survey of the subjects for the evaluation of the level of stress. All subjects initially declared the presence of exam stress, based on their own subjective assessments, and after experimental procedures, they reported a decrease in stress level, and an improvement in their emotional state and mood.
Sincerely – Alexander Fedotchev & co-authors.

Round 2
Reviewer 3 Report (Previous Reviewer 3)
Dear Editors and Authors,
Thanks for inviting me again for the review assignment. After reading the latest version, sorry that I will maintain my recommention to this submission as "rejection".
The reasons can be summarized as follow:
1) even changing to "sign of cognitive improvement", measures regarding exam stress still cant be deemed cognitive assessments. I have suggested the authors to use mood/emotion and remove cognitive rehabilitation from the paper in my previous comments.
2) The assessment for exam stress using 10-point VAS seems too simple and not specific. My concern is, at baseline screening, the subjects need to be well defined as students with (a certain level) exam stress. I can assume not all students during exam period have mental stress. This condition cannot be simply defined as a 10-point self reported test.
3) I wont classify the light system as NIBS. The effect of light on brain remain unclear - highly depending on light property such as intensity. The effect on brain modulation using LED stimulation is not specific like TMS or ultrasonic stimulation.
Author Response
Dear Editors and Reviewer-3,
I see that the reviewer-3 disagrees in principle with some minor details of the work (for example, criticism of future experiments with a 10-point stress rating). Note that his remarks do not diminish the importance of the main results of the study, which informs the readers of "Life" about a development of innovative approach to non-invasive brain stimulation. I can no longer dispute with him - everything has already been said.
Kind regards, Alexander Fedotchev & co-authors.
This manuscript is a resubmission of an earlier submission. The following is a list of the peer review reports and author responses from that submission.
Round 1
Reviewer 1 Report
The article is well written and is related to a very important topic in the field: the improvement of neurostimulation techniques as an important tool for rehabilitation.
Nevertheless, the article needs to be highly improved in order to be considered for publication:
Major revisions:
· -It is not explained the theoretical framework or mechanism relating stress and this proposed neuromodulation technique.
· -Methodology is not well explained and needs to be further detailed. The difference between both techniques is not clear. Questionnaires scores need to be presented.
· -Authors declare in the abstract, results and conclusions that their proposed technique is able to reduce stress in students even when they also recognize that proper stress test was not performed. Therefore, there is not supporting evidence for this asseveration in current version of the manuscript.
-Relative to the assessment of stress levels: how did you measure that those participants with resonance scanning experienced less stress than the other group with EEG-guided adaptive neurostimulation only?
In the conclusions: “In general, our research shows that the developed combined neurostimulation approach could be used in a wide range of stress-reducing and cognitive rehabilitation interventions”
This conclusion is highly subjective and has not proper evidence to be formulated.
· Figures 1 and 2 have not a proper quality (seem to be screenshots from the recording equipment). It is said in the description of the figure that the z-coordinate spectral density (µV/Hz) reflected in the color intensity, but no colors can be discriminated in the figures. Figures should be remake using data recorded.
- In line 119 at the Procedure section, it is stated: “At the end of each experiment, the subjects were retested and interviewed about their feelings during the treatment.”
Authors should assess the effect of retaking the tests in performance, even without stimulation. How can we be sure that the improvement on performance is not related to answering the cognitive test for a second time?
Author Response
Dear Reviewer!
Thank you very much for your thorough and helpful analysis of our paper. We are answering point by point below. Our corrections to the manuscript are shown in yellow.
- 1.-It is not explained the theoretical framework or mechanism relating stress and this proposed neuromodulation technique.
We are trying to correct the corresponding parts of Introduction as follows:
This technique has a number of advantages, such as high personalization of the treatment, automatic operation without conscious efforts of an individual, and integration of interoceptive signals in the regulatory mechanisms of the brain [3].
EEG-guided adaptive neurostimulation has been successfully used for cognitive rehabilitation of stroke patients [4] and for the correction of stress-induced states [5]. The developed technique includes simultaneous presentations of music-like stimuli on-line controlled by the feedback from discrete low-frequency (4-20 Hz) components of the subject’s electroencephalogram – EEG oscillators, and presentations of rhythmic light stimuli generated from the raw EEG of the subject. It was shown that under such bisensory stimulation a decrease of stress level, normalization of the EEG, and positive shifts in the psycho-emotional status of human participants are observed due to the optimal conditions for involving the integrative, adaptive and resonance mechanisms of the central nervous system into human functional state regulation [5]. However, the opportunity to enhance the effectiveness of EEG-guided adaptive neurostimulation is a challenging task and needs further investigation, especially the stress-related line of research. In particular, it is critically important to effectively correct the consequences of stress, which has detrimental effects on psychological well-being, cognitive function, and physiological health of university students [6] and is inversely related with mindfulness and with the results of the most challenging exam [7].
- 2 .Methodology is not well explained and needs to be further detailed. The difference between both techniques is not clear. Questionnaires scores need to be presented.
In order to follow your recommendations, we
1) have changed the third paragraph of Introduction as follows:
The effects of the EEG-guided adaptive neurostimulation significantly depend on the strength of the modulating factor, that is, the EEG of a person. Therefore, one of the possible ways to increase the effectiveness of EEG-guided adaptive neurostimulation may be the preliminary activation of low-frequency rhythmic EEG components of the individual. With this aim, the procedure of resonance activation of the EEG could be used. It consists of LED photostimulation with a stepwise increase in frequency in the range of theta, alpha and beta EEG rhythms, i.e. 4–20 Hz [8]. These EEG rhythms are known to represent several mental conditions, such as deep and wakeful relaxation, good mood, calmness, increased self-awareness and focus, learning of new information, and active attention [9]. Therefore, resonance activation of these rhythms is expected to induce the signs of cognitive rehabilitation of a person. Recently this procedure was successfully used for resonance activation of cortical rhythms in younger schoolchildren [10], where it was named as “resonance scanning” technique.
2) Besides, to present questionnaires scores and to show that two experimental groups of subjects have similar initial scores, we have changed the table 2:
To evaluate the effects obtained as a result of both types of stimulation, the psychophysiological indicators before and after treatment in the experiments with vs without resonance scanning were compared (Table 2).
Table 2.
Objective and subjective indicators before and after stimulation and their levels of difference P in the experiments with vs without resonance scanning (RS).
|
Indicators |
Experimental session |
Before stimulation |
After stimulation |
Р |
|
Theta EEG power (µV2/Hz) |
With RS |
27.3±1.4 |
27.8±1.4 |
0,194 |
|
Without RS |
28.2±1.4 |
28,6±1.4 |
0,166 |
|
|
Alpha EEG power (µV2/Hz) |
With RS |
28.3±1.9 |
31.9±2.1 |
0,001 |
|
Without RS |
28.0±1.8 |
28.6±1.7 |
0,095 |
|
|
Beta EEG power (µV2/Hz) |
With RS |
12.8±0.9 |
13.7±0.8 |
0,007 |
|
Without RS |
14.3±0.9 |
14.5±1.0 |
0,361 |
|
|
Errors in word recognition test (digits) |
With RS |
4.4±0.5 |
2.5±0.3 |
0,002 |
|
Without RS |
4.2±0.4 |
3.6±0.5 |
0,207 |
|
|
Level of emotional maladjustment (scores) |
With RS |
4.6±0.4 |
2.8±0.3 |
0,008 |
|
Without RS |
4.3±0.8 |
3.8±0.8 |
0,571 |
Note: the values of changes with the significance level P <0,05 are shown in bold (Mann–Whitney test)..
- Authors declare in the abstract, results and conclusions that their proposed technique is able to reduce stress in students even when they also recognize that proper stress test was not performed. Therefore, there is not supporting evidence for this asseveration in current version of the manuscript. 4. Relative to the assessment of stress levels: how did you measure that those participants with resonance scanning experienced less stress than the other group with EEG-guided adaptive neurostimulation only?
It is the main limitation of our study. Unfortunately, we used only a comparison of data from the initial and final survey of subjects on the level of stress, instead of quantitative assessments of stress level. All subjects initially declared the presence of examination stress, based on subjective assessments, and after experimental procedures they reported a decrease in the level of stress. In future research, this limitation will be overcome by developing reliable means of assessing the level of stress or using a 10-point visual analog scaling technique. Therefore, we removed any mention of stress reduction effects from the abstract, results, and conclusions.
For example, in the manuscript it will be: “Thus, resonance scanning could be used as a tool to enhance the efficiency of EEG-guided adaptive neurostimulation in cognitive rehabilitation procedures.” instead of “Thus, resonance scanning could be used as a tool to enhance the efficiency of EEG-guided adaptive neurostimulation in stress-reducing and cognitive rehabilitation procedures.”
- In the conclusions: “In general, our research shows that the developed combined neurostimulation approach could be used in a wide range of stress-reducing and cognitive rehabilitation interventions. This conclusion is highly subjective and has not proper evidence to be formulated.
We have corrected this phrase as follows:
In general, our research shows that the developed combined neurostimulation approach after the thorough elaboration could be used in a wide range of cognitive rehabilitation interventions.
- 6. Figures 1 and 2 have not a proper quality (seem to be screenshots from the recording equipment). It is said in the description of the figure that the z-coordinate spectral density (µV/Hz) reflected in the color intensity, but no colors can be discriminated in the figures. Figures should be remake using data recorded.
Yes, both Figures are screenshots from the recording equipment. It was our intention to demonstrate to the readers the dynamics of resonant EEG reactions at the stimulation frequency and the rhythm multiplication EEG reactions during the procedure of resonance scanning. In order to follow your recommendation, we have changed these two figures as follows:
Fig. 1. Dynamics of the stimulation (left) and the EEG (right) spectra during the experiment with subject R08.
X-axis - the frequency of the spectrum, Hz;
Y-axis - the time of the experiment, sec.
Z-axis is the spectral density (µV/Hz) reflected in the level of brightness.
Fig. 2. Dynamics of the stimulation (left) and the EEG (right) spectra during the experiment with subject E12. Designations as in Fig. 1.
- - In line 119 at the Procedure section, it is stated: “At the end of each experiment, the subjects were retested and interviewed about their feelings during the treatment.”8. Authors should assess the effect of retaking the tests in performance, even without stimulation. How can we be sure that the improvement on performance is not related to answering the cognitive test for a second time?
In the corrected version of Table 2 we have presented the data from performance tests before and after stimulation (see also the answer № 2 above)
Table 2.
Objective and subjective indicators before and after stimulation and their levels of difference P in the experiments with vs without resonance scanning (RS).
|
Indicators |
Experimental session |
Before stimulation |
After stimulation |
Р |
|
Theta EEG power (µV2/Hz) |
With RS |
27.3±1.4 |
27.8±1.4 |
0,194 |
|
Without RS |
28.2±1.4 |
28,6±1.4 |
0,166 |
|
|
Alpha EEG power (µV2/Hz) |
With RS |
28.3±1.9 |
31.9±2.1 |
0,001 |
|
Without RS |
28.0±1.8 |
28.6±1.7 |
0,095 |
|
|
Beta EEG power (µV2/Hz) |
With RS |
12.8±0.9 |
13.7±0.8 |
0,007 |
|
Without RS |
14.3±0.9 |
14.5±1.0 |
0,361 |
|
|
Errors in word recognition test (digits) |
With RS |
4.4±0.5 |
2.5±0.3 |
0,002 |
|
Without RS |
4.2±0.4 |
3.6±0.5 |
0,207 |
|
|
Level of emotional maladjustment (scores) |
With RS |
4.6±0.4 |
2.8±0.3 |
0,008 |
|
Without RS |
4.3±0.8 |
3.8±0.8 |
0,571 |
These data show that before both types of stimulation the cognitive tests demonstrate similar scores. Although both group of subjects received the same posttreatment testing, only after the combined stimulation (with resonance scanning) there were significant reductions in recognition errors and in the level of emotional maladjustment.
Sincerely – Alexander Fedotchev & co-authors.

Reviewer 2 Report
In this work the Authors applied resonance scanning and EEG-guided adaptive stimulation to one group of students under exam stress and compared the EEG power and the psychophysiological state of this group with a matched control group receiving only EEG-guided adaptive stimulation without resonance scanning.
I found this work interesting and the overall methodology and results are clear.
However, I suggest some minor changes and ask for some clarifications:
1. the composition of the two groups is not clear to me, I suggest to add a table reporting the subjects, their gender, their age if you collected it, and the group in which they belong.
2. In data analysis there is a mention of the Shapiro-Wilk test, in my opinion the p-value of this test must be reported somewhere, if I did not miss it.
3. Also, in Data analysis you use ANOVA (a parametric test) and then a Mann–Whitney Rank Sum Test (a non parametric test), and then stating "The paired t-test was used to determine the mean values (M) and standard errors (m)". Which test was used? Mann-Whitney rank sum (non parametric) or paired t-test (parametric)? In this case I personally prefer the Mann-Whitney, as the numerosity of your data sample is still small, but with a negative (p<0.05) Shapiro Wilk test you could use both in my opinion.
4. To help the reader I suggest to add the statistical test name on Figure 3 and Table 1.
5. A typo I think, check P values of table 1 for "," instead of "."
Author Response
Dear Reviewer!
Thank you very much for your kind and helpful comments. We are answering step by step below. Our corrections to the manuscript are shown in yellow.
- the composition of the two groups is not clear to me, I suggest to add a table reporting the subjects, their gender, their age if you collected it, and the group in which they belong.
We added such table as follows:
The experiments involved two groups of healthy students from Lobachevsky State University of Nizhni Novgorod. A number of participants, age, gender and study background of members were practically equal (Table 1).
Table 1.
Composition of two groups of university students participated in the experiments with or without resonance scanning.
|
Group attribute |
Group |
|
|
With resonance scanning |
Without resonance scanning |
|
|
Number of participants |
18 |
18 |
|
Gender (females/males) |
11/7 |
11/7 |
|
Age (years) |
20.7±0.2 |
21.0±0.2 |
|
Years of study |
3-4 |
3-4 |
2 In data analysis there is a mention of the Shapiro-Wilk test, in my opinion the p-value of this test must be reported somewhere, if I did not miss it. 3. Also, in Data analysis you use ANOVA (a parametric test) and then a Mann–Whitney Rank Sum Test (a non parametric test), and then stating "The paired t-test was used to determine the mean values (M) and standard errors (m)". Which test was used? Mann-Whitney rank sum (non parametric) or paired t-test (parametric)? In this case I personally prefer the Mann-Whitney, as the numerosity of your data sample is still small, but with a negative (p<0.05) Shapiro Wilk test you could use both in my opinion.
We have corrected this paragraph as follows:
- Data Analysis
A statistical data analysis was performed by using the software package Sigma-Plot 11.0.
To check the normality of the data distribution, the Shapiro-Wilk W-test was used, since this test has the highest power and is most preferable for small samples. The analyzed distribution was assessed as normal at the level of statistical significance p>0,05. In the case of normal distribution of data for descriptive statistics, the mean (M) and standard errors (m) were used; to assess the significance of differences, Student's t-test was used. In the case of a distribution other than normal, the median (Me) and the range of values from the first (Q1) to the third (Q3) quartile were used for descriptive statistics of features, and the Mann–Whitney test was used to assess the significance of differences. The critical level of statistical significance was taken as p<0,05. Repeated measures multiway ANOVA was used to assess the statistical significance of the shifts for each psychophysiological indicator under stimulation relative to the background. Power shifts for theta, alpha, beta EEG rhythms, changes in a number of errors in the word recognition test, and the shifts of the level of emotional maladjustment were processed.
- To help the reader I suggest to add the statistical test name on Figure 3 and Table 1.
The test name (Mann-Whitney test) is added to the Notes for Figure 3 and Table 2 as follows.
Fig. 3. EEG theta-, alpha-, beta-rhythm powers before (dark columns) and during (light columns) resonance scanning. Note. * - P< 0,05 (Mann–Whitney test).
Table Note: the values of changes with the significance level P <0,05 are shown in bold (Mann–Whitney test).
- A typo I think, check P values of table 1 for "," instead of "."
Thank you, we corrected it.
Sincerely – Alexander Fedotchev & co-authors.

Reviewer 3 Report
Fedotchev et al., demonstrated a close-loop stimulation system using LED stimulation to modulate the cortical EEG. It showed that the system can increase the cortical rhythmic powers and improve the performance in cognitive tasks and emotions of students with mental stress due to examinations.
The topic is interesting; however, I feel that the authors did not well describe the experimental details, including inclusion/exclusion criteria, behavioral tasks, etc. Furthermore, the implication of the study is not clear. As a reader, I cannot understand why the system can improve behavioral outcomes in students by using LED stimulation.
I have a few point-by-point comments for the authors to consider:
Simple summary:
Non-drug should be non-pharmacological
Abstract:
The current design seems to me is an open-loop: sensory stimulation modulates the EEG. EEG does not provide feedback to external perturbation.
Number of participants should be mentioned in the abstract: How many university students were enrolled?
The implication may not be “cognitive rehabilitation”. It focuses on stress management which is more emotional or in the field of mental health.
Introduction
Page 80:
Beta is usually 12-30Hz. So theta-beta is 4-20 Hz is not true.
Pages 88-89: Again, I do not agree that the current system is designed for cognitive rehabilitation.
Methods:
Page 101:
“practically equal” is confusing. Just say “non-significant difference at baseline”
Table 1: No statistical comparison on baselined demographics.
Any randomization for separating the 2 groups?
Page 136: Remove %
Which ear was used as reference? Where was the ground sensor?
Results:
Figure 1: Pls remove the background color.
Fig 2: The data collected at resting eye close? (alpha is high). Need to mention the experimental details in the methods.
Table 2: It seems to me the authors did not mention the methodological details of the 2 behavioral tasks: word recognition test and emotional maladjustment.